# Gene‒Prostate-Specific-Antigen-Guided Personalized Screening for Prostate Cancer

**DOI:** 10.3390/genes10090641

**Published:** 2019-08-24

**Authors:** Teng-Kai Yang, Pi-Chun Chuang, Amy Ming-Fang Yen, Hsiu-Hsi Chen, Sam Li-Sheng Chen

**Affiliations:** 1Division of Urology, Department of Surgery, Yonghe Cardinal Hospital, New Taipei 23445, Taiwan; 2Institute of Epidemiology and Preventive Medicine, College of Public Health, National Taiwan University, Taipei 10055, Taiwan; 3School of Medicine, College of Medicine, Fu Jen Catholic University, New Taipei 24205, Taiwan; 4Medical Education Department, Far Eastern Memory Hospital, New Taipei 22060, Taiwan; 5School of Oral Hygiene, College of Oral Medicine, Taipei Medical University, Taipei 11031, Taiwan

**Keywords:** effectiveness, prostate cancer, risk stratification, screening

## Abstract

(1) Background: A simulation approach for prostate cancer (PrCa) with a prostate-specific antigen (PSA) test incorporating genetic information provides a new avenue for the development of personalized screening for PrCa. Going by the evidence-based principle, we use the simulation method to evaluate the effectiveness of mortality reduction resulting from PSA screening and its utilization using a personalized screening regime as opposed to a universal screening program. (2) Methods: A six-state (normal, over-detected, low-grade, and high-grade PrCa in pre-clinical phase, and low-grade and high-grade PrCa in clinical phase) Markov model with genetic and PSA information was developed after a systematic review of genetic variant studies and dose-dependent PSA studies. This gene‒PSA-guided model was used for personalized risk assessment and risk stratification. A computer-based simulated randomized controlled trial was designed to estimate the reduction of mortality achieved by three different screening methods, personalized screening, universal screening, and a non-screening group. (3) Results: The effectiveness of PrCa mortality reduction for a personalized screening program compared to a non-screening group (22% (9%‒33%)) was similar to that noted in the universal screening group (20% (7%‒21%). However, a personalized screening program could dispense with 26% of unnecessary PSA testing, and avoid over-detection by 2%. (4) Conclusions: Gene‒PSA-guided personalized screening for PrCa leads to fewer unnecessary PSA tests without compromising the benefits of mortality reduction (as happens with the universal screening program).

## 1. Introduction

Although population-based screening for prostate cancer (PrCa) with prostate-specific antigen (PSA) testing reduces mortality from PrCa by 20% according to the European Randomized Study of Screening for Prostate Cancer (ERSPC) after a long-term follow-up [1], a large proportion of over-detection and over-treatment of PrCa resulting from the PSA test leaves the use of PSA as a population-based screening method with much to be desired [2,3]. 

The previous studies on the validity of PSA as a screening test for PrCa have already shown that there is no single cutoff of PSA that can attain the performance of “ruling in” and “ruling out” criteria [4,5,6]. Over the past three decades, there are numerous studies on genetic variants that have identified a constellation of SNPs accounting for the risk of being susceptible to PrCa [7]. PSA-associated markers have also been identified by genome-wide association studies [8,9,10]. Information from genetic markers might further enhance the utility of PSA testing. Gudmundsson et al. have identified such PSA-associated markers in a genome-wide association study to propose a personalized PSA cutoff value based on genotype [11]. The combination of PSA with SNPs in a Finnish study to predict the risk of PrCa shows better performance, enhancing the area under the receiver operating characteristic curve from 70% to 96% in men with a PSA level higher than 4 ng/mL [12]. 

While a combination of genetics and the PSA test is useful for predicting the risk of PrCa, very few studies have combined the information on PSA and genes into a multistate disease natural history from the pre-clinical detectable phase (PCDP) to the clinical phase (CP), which is generally modeled by using the data available from various detection modes of mass screening [2,13,14]. Doing so enables one to stratify the underlying population into different risk groups, which provides the basis for the development of personalized screening for PrCa, changing the screening interval and age to begin screening according to the risk profiles. A previous study has shown how to use genetic variants associated with PrCa to develop a personalized screening strategy [12]. However, it has not taken PSA into account. Recent studies have found a dose-dependent effect of PSA on the incidence of PrCa [15,16]. This prompted us to extend the gene-guided model into a gene‒PSA-guided natural history model of PrCa, which could provide more precise screening strategies for PrCa as a result of combining the information. It was postulated that the application of a personalized screening strategy could reduce the use of PSA tests, particularly for population-based screening. It is therefore interesting to evaluate the effectiveness and the utilization of PSA testing in the context of personalized screening in comparison with universal screening. 

The aims of this study were to develop a gene‒PSA-guided multistate disease natural history model of PrCa, making allowances for over-detection, on which a computer-based simulated randomized controlled trial was designed and conducted to evaluate the effectiveness of reducing mortality from advanced PrCa, and to assess the utilization of PSA testing in personalized screening programs in comparison with the universal screening program. 

## 2. Materials and Methods

### 2.1. Study Design

A computer-based randomized controlled trial design was proposed to simulate a population-based trial with three groups, a personalized screening group (PSG), a universal screening group (USG), and a non-screening group (NSG). The three groups were generated by using a computer simulation technique developed in a previous study [14] to generate a cascade of phenotypes over time (from normal to PCDP to CP) by following the disease’s natural history model superimposed with information on PSA level and genetic variants, as described below. Based on this natural history model, two intervention arms were created to interrupt the disease’s natural history so as to increase early detection and treatment and allow a better survival rate. Those who are assigned to the PSG were offered different screening intervals and ages to begin screening in the light of a risk stratification based on the PSA level and genetic score, composed of the 20 SNPs specified below. The USG adopted a universal policy of four-yearly screenings provided to men aged between 55 and 75 years. The NSG received no screening. 

Following a previous method of sample size determination for a population-based randomized controlled trial [17], we estimated the sample size required for surrogate endpoint such as late cancer (Gleason 7+) and for primary endpoint of mortality, assuming the scenario and the parameters (the incidence of PDCP, case-fatality rate, the distribution of advanced cancer, and the duration of study) were identical to the ERSPC trial. Given 5% type I error, each arm may require at least 6000 men for the surrogate endpoint and at least 15,000 men for the primary endpoint in order to meet 80% statistical power. Note that as the PSG contained information on PSA and genetic variants, the required sample size was further reduced as more information on genetics and PSA was available. 

### 2.2. The Natural History Model of PrCa with PSA and Genetic Variants

In order to generate the control group, i.e., the NSG in the absence of screening, the six-state (normal, over-detected, low-grade and high-grade PrCa in pre-clinical phase, and low-grade and high-grade PrCa in clinical phase) Markov model for disease natural history combined with PSA level and genetic variants was developed. Figure 1 shows the gene‒PSA-based six-state model for non-progression and progression of low-grade and high-grade PrCa from normal, PCDP, and to CP. SNPs associated with the incidence and aggressiveness of prostate cancer are listed in Table 1. The selection of these SNPs reside in genomic locations that, based on previous findings, show not only significant risk effects but also have common and consistent association with prostate cancer susceptibility in various populations. For instance, the rs4242382, rs16901979, rs6983267, rs1447295 at 8q24 are the common and consistent risk SNPs for developing prostate cancer [12,18,19,20]. The rs138213197 in HOXB13 at 17q22 has been shown to be associated with the prostate cancer not only in the Caucasians in the US but in a Finnish population [21,22]. Moreover, considering the availability of empirical genetic information for distinguishing the initiators and promoters, and non-aggressive and aggressive prostate cancer, we selected these genetic variants for our natural history model. The transition parameters combining with the information on the incremental effect of PSA on the occurrence of preclinical PrCa with and without progression were further tuned in the light of previous studies proving the effect of PSA on incidence in a dose‒response manner after systematic review [12,15,16]. We applied the meta-analysis of the Poisson regression model with a random-effect term to capture the heterogeneity across three studies from Finland, the USA, and Denmark. The PSA level was categorized into eight groups: ≤1, 1‒2, 2‒3, 3‒4, 4‒6, 6‒8, 8‒10, and >10 ng/mL. For the Finnish trial, we extracted the number of cancers and person-years of the eight groups. For the USA and Denmark studies, a wider range of some given PSA levels, for example, 3‒10 ng/mL, were given different weights of dummy variables according to the proportion of detailed groups in the Finnish trial. Table 1 also lists the clinical weights of SNPs and PSA levels corresponding to the incidence and aggressiveness of prostate cancer. The clinical weights were derived from the logarithm transform of relative risk shown in the literature/meta-analysis. Finally, the risk scores were determined by the combination of risk profile times the corresponding clinical weights for each man. The formulae for the two risk scores for the incidence and aggressiveness of prostate cancer, respectively are shown in the Appendix A. 

### 2.3. Risk Stratification

Information was fed into the disease natural history model by dividing the PSA level into eight groups, as mentioned above. These were further divided into two categories, high and low risk group, according to the gene score from SNPs. 

### 2.4. Computer Simulation

We developed a micro-simulation computer algorithm of a three-arm randomized controlled trial design with 15,000 men in each arm. Every man started from PrCa-free at age 40 and followed the natural history year by year to determine his status of disease according to the annual transition probabilities, which were determined by the personal genetic risk profiles and PSA level. The numbers of low-grade and high-grade PrCa and deaths from PrCa or other causes of death between ages 40 and 85 were estimated. In the USG, PSA screening was provided from the age of 55 up to 75, every four years. In the year of screening, PrCa in PCDP was picked up as screen-detected cases with consideration of the sensitivity of the PSA testing. Otherwise, PrCa in PCDP could further progress naturally to CP or stay in the PCDP until being picked up by the next round of screening. Men of different risk groups were offered different screening strategies with a different starting age of screening and between-screenings interval (see below). We used the ratio of the number of high-grade PrC/PrC deaths of USG/PSG to those of NSG to measure the effectiveness of the screening. Note that the number of cases that occurred before age 55 were removed in order not to dilute the true effectiveness of screening. 

## 3. Results

### 3.1. The Effect of PSA on the Risk of Developing Prostate Cancer

A meta-analysis was performed to estimate the dose‒response effect on the risk of developing PrCa, as shown in Table 1. Compared to those with a PSA level below 1.0 ng/mL, the estimated relative risk increased from 3.82 (95% CI: 2.97‒4.91) in men with a PSA level of 1.01‒2.0 to 115.8 (95% CI: 84.7‒153.6) in men with PSA > 10 ng/mL. The risk of different PSA levels was then logarithmically transferred to the weights from 1.3415 at a lower PSA level to 4.7430 at a high PSA level to form the risk score on the incidence of prostate cancer.

### 3.2. Estimates of the Risks for PrCa by Genetics and PSA Level 

Based on PSA and genetic information, the population was stratified into 16 risk groups. Table 2 shows the 10-year risk for prostate cancer, as well as the positive and negative likelihood ratios by PSA levels, given the genetic risk information. The 10-year risk for prostate cancer increased from 0.3% in the lowest-risk group (PSA ≤ 1 ng/mL and lower gene score) to 72.5% in the highest-risk group (PSA > 10 ng/mL and higher gene score). The positive likelihood ratios based on the risk stratification ranged from 1.30 for the lowest-risk group to 69.2 for the highest-risk group. The corresponding negative likelihood ratio ranged from 0.08 to 0.93. 

The positive likelihood ratio exceeded 10 for men with PSA > 8 ng/mL or those with PSA between 4.01 and 8 ng/mL but with a high genetic risk profile. In addition to PSA, genetic information was helpful in determining the risk of developing prostate cancer. For instance, if the genetic risk profile was high, a high positive likelihood ratio of 7.7 can be reckoned even in men with PSA between 3 and 4 ng/mL.

### 3.3. Personalized Screening Policy

Table 3 presents the corresponding results for the recommended age for starting screening and the screening interval for different risk groups. For the medium-risk group, a screening policy commencing at 55 years old with a four-year screening interval was suggested. For the men with the highest risk (the two highest-risk groups with PSA > 8 ng/mL and with a high genetic risk profile of LR+ larger than 39), the recommended age at which to begin screening was 47 years old, and a more intensive one-year interval was recommended. For the low-risk group, the starting age was postponed to 60 years old and the screening interval was lengthened to 12 years (negative likelihood ratio less than 0.18). 

### 3.4. The Effectiveness of Universal and Gene‒PSA-Guided Personalized Screening Regimes 

Compared with NSG, USG with a four-year screening interval could reduce mortality from PrCa by 20%. PSG could reduce PrCa mortality by 22% (Table 4). In terms of the effectiveness of high-grade PrCa reduction, USG and PSG could achieve 37.0% and 41.3%, respectively, when compared with NSG. The PSG resulted in 2% less detection of non-progressive PrCa (over-detection) than USG. The USG and PSG accumulated 88,673 and 65,586 PSA tests, respectively, yielding a reduction of PSA utilization by 26% if PSG, instead of USG, was adopted.

## 4. Discussion

### 4.1. Benefits of the Gene‒PSA-Guided Personalized Screening Regime 

This was the first study to build a gene‒PSA-guided multistate natural history of PrCa, on which we developed a personalized screening regime in terms of the screening interval and the age at which to begin screening. We then evaluated the effectiveness of PSA testing in a personalized screening regime in comparison with a universal screening regime, as seen in the majority of the ERSPC trial. 

There are two main reasons for adopting a personalized screening regime. It not only reduces false negative PrCa in the high-risk group by shortening the screening interval and beginning the screening at an earlier age, but also avoids false positive PrCa and even extreme cases of over-detection by lengthening the screening interval and setting the age to start screening later in the low-risk group. As the majority of the underlying population lies below the average risk group, periodical PSA testing with a longer screening interval was preferred. This could account for why a personalized screening regime could eliminate approximately one-quarter of PSA tests in comparison with a universal screening regime. In this sense, it is conceivable that a personalized screening regime may lead to a better quality of life in terms of false negative PrCa leading to aggressive treatment of these missed cancers and also false positive cases and over-detected PrCa requiring unnecessary biopsy and confirmatory procedures. On the other hand, the cost saving from reducing unnecessary PSA tests, follow-up biopsy, and confirmatory procedures can partially cover the extra costs for genetic tests in the personalized screening group. Nonetheless, the gene‒PSA-guided personalized screening strategy did not require the whole population to undergo genetic testing at a high cost. Instead, as we found in our example, the genetic risk score played a minor role in distinguishing risk for men with PSA below 1.0 ng/mL, which covers half of men. Namely, the PSA‒gene-guided approach can cut the costs of genetic testing in half compared with the gene-guided approach. 

### 4.2. Personalized Multistate Risk Assessment Model with Gene and PSA

The contribution of this study was to translate the findings on SNPs in combination with PSA level into practical risk stratification, which has never been addressed before. There are two advantages to this individual risk assessment. First, risk stratification of the underlying population with PSA and genetics can prioritize intervention targets. Second, 10-year and lifetime risk prediction are useful for providing genetic counseling for patients who undergo genetic and PSA testing. Doing so is a great aid to shared decision-making between physicians and patients. In the example of PrCa, the high-risk group and the low-risk group can both benefit from the personalized risk assessment model as those at high risk can be given aggressive treatment and therapy, whereas those at low risk can follow a policy of watchful waiting. 

### 4.3. Economical Aspects of Genetic Testing 

Since genetics in combination with PSA makes a contribution to the multistate disease natural history of PrCa, the costs of genetic testing play a crucial role in the cost-effectiveness of the gene‒PSA-guided personalized screening regime. A threshold test of cost-effectiveness analysis should be conducted, considering different levels of willingness to pay or gross domestic product in order to estimate the optimal economic scale of genetic testing. This means that the costs of genetic testing should fall if the number of genetic tests increases. 

### 4.4. Clinical Relevance of Computer Algorithms Considering Personalized Risk 

The development of our clinical algorithm can facilitate shared decision-making for clinicians and patients by providing information from the personalized risk model—for example, charts showing the cumulative risks of high-grade PrCa and PrCa death for patients with different PSA levels and genetic risk (Appendix A). Patients in a specific risk group can decide the best screening strategy. For example, those with a high level of PSA, say 10 ng/mL, and high genetic risk may find that the personalized screening policy results in a reduction of high-grade PrCa and PrCa death risk. In contrast, those at lower risk may choose a less aggressive screening policy.

### 4.5. Limitations 

There were some limitations to the current study. First, in addition to genes and PSA, there may be other factors, including undiscovered genes, particularly in association with lower risk, that can be used for refining the risk stratification. The proposed gene‒PSA-guided natural history model should keep pace with the updated knowledge. Second, the current model was based on state-of-art evidence on genetic and PSA effects by including a number of ethnic groups, such as Finnish, Danish, and Latino and non-Latino subjects in the United States. Whether the application of this model to specific ethnic groups is appropriate may need further attention. Finally, the Markov property has been applied to six-state disease natural history, but it may not be fully applicable to prostate cancer. In addition, to develop a non-progressive state for accommodating over-detection, to be amenable to non-Markov property, an exponential regression model was used to incorporate PSA and genes to turn a non-Markov process into a Markov process. The alternative would have been to use a semi-Markov model for accommodating a non-Markov process. This is the subject of ongoing research. 

There are now over 100 known hereditary PrCa-associated SNPs, which can be incorporated in the risk score. With the increase in the number of SNPs, a better understanding of the role of this genetic information together with PSA on the natural history of prostate cancer will be very useful for risk stratification to guide the screening policy for prostate cancer. More research is needed to verify the developed approach.

In conclusion, we built a gene‒PSA-guided multistate disease natural history model of PrCa and applied it to develop a personalized screening regime with various screening intervals and ages to begin screening, which could dispense with unnecessary PSA testing for those at low risk, without compromising the benefits of mortality reduction in the population (as happens with the universal screening program).

## Figures and Tables

**Figure 1 genes-10-00641-f001:**
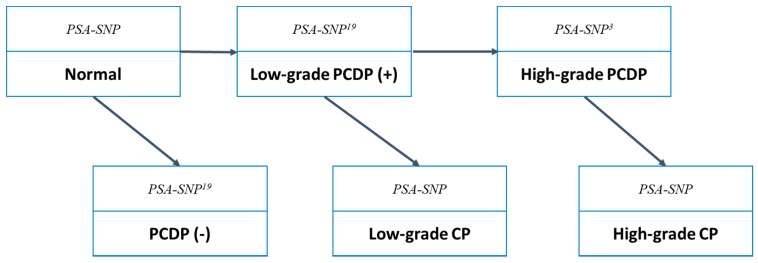
Gene‒prostate-specific-antigen (PSA)-based six-state model for non-progression and progression of low-grade and high-grade prostate cancer (PrCa). The superscript numbers beside SNP indicate the number of SNPs used in the transition risk. CP: clinical phase; PCDP: preclinical detectable phase.

**Table 1 genes-10-00641-t001:** The distribution in population and weights for the risk of incidence and aggressiveness of prostate cancer by PSA levels, selected SNPs, and family history.

**Prostate-Specific Antigen, SNPs, and Family History**			**% in Population**	**Weights**	**References**
**Incidence of Preclinical Low-Grade PrCa**					
**PSA, ng/mL**					
≤1.0			47.8%	0.0000	[12,15,16]
1.01‒2.0			30.3%	1.3415	
2.01‒3.0			10.7%	2.2147	
3.01‒4.0			5.1%	2.5799	
4.01‒6.0			3.5%	2.7377	
6.01‒8.0			1.3%	3.2789	
8.01‒10			0.5%	3.9098	
>10			0.8%	4.7430	
**SNPs**	**Position**	**Associated Allele**			
rs4242382	8q24 (region)	AA	4.36%	0.4978	[14]
		GA	30.6%	0.1084	
rs138213197	17q21-22	T	1%	3·60	[14]
rs4430796	17q12	TT (30%)	56%	0.3221	[18]
rs1859962	17q24.3	GG (25%)	50%	0.2468	
rs16901979	8q24(region 2)	AA/CA (7%)	3%	0.4253	[18,19,20]
rs6983267	8q24(region 3)	GT/GG (77%)	51%	0.3184	
rs1447295	8q24(region 1)	CA/AA (26%)	14%	0.1988	
rs2660753	3p12	C	11%	0.0769	[23]
rs9364554	6q25	C	28%	0.1310	
rs6465657	7q21	T	47%	0.1132	
rs10993994	10q11	C	39%	0.2231	
rs7931342	11q13	G	50%	‒0.1625	
rs2735839	19q13	G	15%	‒0.1165	
rs5945619	Xp11	T	35%	0.2546	
rs721048	2p15	A	19%	0.3197	[24]
rs5945572	Xp11	A	35.1%	0.2070	
rs10486567	JAZF1 (7)	GG	59.29%	‒0.3011	[25]
		GA	35.42%	‒0.3424	
rs4054823	17p12	T	72%	0.1823	[26]
rs7920517	10	AG	47.6%	0.1988	[27]
**Family history**			4.6%	0.6471	[14]
**From Low-Grade Preclinical PrCa to High-Grade PrCa or Clinical PrCa**					
**SNPs**	**Position**	**Associated Allele**			
rs200331695	11q13	A	0.2%	2.0643	[14]
IGF-I	Q_2_			1.1631	[28]
	Q_3_			1.2528	
	Q_4_			1.6292	
GSTP1 hypermethylation	11:67584109-6758428		68%	1.5151	[29]

**Table 2 genes-10-00641-t002:** The 10-year risk of developing prostate cancer, with the positive and negative likelihood ratios by PSA levels and genetic risk groups.

Prostate-Specific Antigen (ng/mL)	Genetic Risk	10-Year Risk for Prostate Cancer	Positive Likelihood Ratio	Negative Likelihood Ratio
>10	High	72.5%	—	0.93
>10	Low	30.9%	69.17	0.87
8.01‒10	High	43.0%	25.59	0.89
8.01‒10	Low	15.0%	39.23	0.74
6.01‒8	High	27.3%	10.12	0.82
6.01‒8	Low	8.1%	18.06	0.50
4.01‒6	High	17.4%	5.26	0.75
4.01‒6	Low	4.9%	10.65	0.48
3.01‒4.0	High	15.0%	4.71	0.65
3.01‒4.0	Low	4.2%	7.71	0.28
2.01‒3.0	High	10.7%	2.69	0.51
2.01‒3.0	Low	2.9%	5.45	0.24
1.01‒2.0	High	4.7%	2.40	0.31
1.01‒2.0	Low	1.2%	2.83	0.08
0‒1.0	High	1.3%	1.30	0.18
0‒1.0	Low	0.3%	1.55	—

**Table 3 genes-10-00641-t003:** The recommended age to start screening and the screening interval by PSA level and combined genetic risk among subjects susceptible to progressive PCa.

Prostate-Specific Antigen, ng/mL	Genetic Risk	Screening Starting Age, Years	Screening Interval, Years
>10	High	47	1
>10	Low	50	2
8.01‒10	High	47	1
8.01‒10	Low	52	3
6.01‒8	High	50	2
6.01‒8	Low	55	4
4.01‒6	High	52	3
4.01‒6	Low	55	4
3.01‒4.0	High	52	3
3.01‒4.0	Low	55	4
2.01‒3.0	High	52	3
2.01‒3.0	Low	55	4
1.01‒2.0	High	55	4
1.01‒2.0	Low	60	12
0‒1.0	High	60	12
0‒1.0	Low	60	12

**Table 4 genes-10-00641-t004:** Simulated results of no screening, universal, and gene‒PSA personalized prostate cancer screening.

	NSG	USG	PSG
**Participants**	15,000	15,000	15,000
**Prostate cancer deaths, *n***	384	307	299
**Mortality reduction, rate ratio (95% CI)**	Reference	0.80	0.78
		(0.67‒0.91)	(0.69‒0.93)
**High-grade cancers, *n***	251	158	148
**Incidence reduction, rate ratio (95% CI)**	Reference	0.63	0.59
		(0.52‒0.77)	(0.48‒0.72)
**Number of PSA tests, *n***	-	88,673	65,586
**Test reduction, %**	-	Reference	26
**Over-detection cases, *n***	-	193	190
**% of avoid over-detection**	-	Reference	2

NSG: non-screening group; USG: universal screening group; PSG: personalized screening group.

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
