# Peer review of "Gene‒Prostate-Specific-Antigen-Guided Personalized Screening for Prostate Cancer"

_genes, 2019, doi:10.3390/genes10090641_

Round 1
Reviewer 1 Report
General
1. The manuscript should be thoroughly edited for grammar and English language usage. In several instances, I was unable to comprehend the authors’ messaging on account of sentence structure and word choice. I worry that I was unable to review effectively for lack of understanding of the language.
2. At no point do the authors discuss the work of Gudmundsson, et al. (Sci Transl Med 2010), which was the first investigation of PSA adjusted for genetics. And have there been any such studies since? I would recommend that the authors complete a thorough search for existing relevant literature.
3. Given that the simulated model presumably represented “ideal” conditions, would the term “efficacy” be more appropriate than “effectiveness”? Or perhaps I’m unclear about study design? Regardless, the terminology should be consistent throughout.
Abstract
4. What do the authors mean by “multistate disease natural history”? Because the first sentence is unclear, the motivation for the study does not come through.
5. It is unreasonable to state that personalized screening reduced death from PrCa relative to universal screening. The confidence intervals have substantial overlap. The authors need to tone down their conclusions.
Introduction
6. At the beginning of the second paragraph, the authors refer to a “previous study” without providing a reference.
7. It should not be stated as fact that population-based screening should not yet be introduced. There exist conflicting opinions on the matter. In addition, a 2009 study is not a valid reference when discussing a question for which substantial data have been generated over time.
Methods
8. The authors speak to staging as early and late. In the context of PrCa, staging typically refers to TNM stage. Can they offer alternative terminology?
9. It’s unclear how or why the authors selected 20 SNPs. Over 100 PrCa SNPs have now been identified.
10. What does it mean that the “simulation is on the basis of one year cycle”?
Discussion
11. The authors indicate “so many benefits” from individual risk assessment, but describe only two.
Reviewer 2 Report
Yang et al in this manuscript titled "Gene-Prostate Specific Antigen-Guided Personalized 2 Screening for Prostate Cancer" have described gene-PSA-guided personalized screening for Prostate cancer. The manuscript is almost well written and performed in an organized manner. The manuscript has several implications and indeed would add value to the current set of information. The caveats observed here are the following which the authors may need to take care of:
1) the authors have not explained how the genome-wide association studies have impacted variants within the PSA gene influencing PSA expression in other racial cohorts of men.
2) The study also does not explain in detail the clinical utilities od PSA-SNPs and how it improves PSA screening.
3) What is the importance of clinical algorithms and they could be incorporated with genetic testing to improve current screening and treatments? The authors have described the computer simulations in the methodology but have not actually brought out its actual benefits in the discussion in a visible manner.
4)The authors have provided some limitations but have not suggested any future directions.
5) The authors have only used one ethnic group. What explanation do they provide for other groups?
Round 2
Reviewer 1 Report
1. The manuscript requires further editing for grammar and English language usage. I’m still finding it somewhat difficult to provide a comprehensive review.
2. The updated sentence “PSA-associated markers have been identified by genome-wide association studies [6-9]” does not include the appropriate references. First, it references studies that are not GWAS, and second, it does not include the most recent PSA GWAS available (PMID 28139693). I had also meant to convey that the work of Gudmundsson, et al. should be discussed for its analyses of genetically-adjusted PSA levels.
3. The abstract is still written as though personalized screening definitively outperformed universal screening. In fact, the point estimates are very similar and the confidence intervals have substantial overlap. The authors still need to rework how they describe their results (throughout the manuscript, now that I’ve read the rest).
4. It’s inaccurate to say that there is only one previous study that has evaluated “the validity of PSA as a screening test for PrCa.”
5. Given that the authors are categorizing PrCa according to Gleason grade, the terminology should be high-grade or low-grade rather than early or late stage.
6. I didn’t mean to suggest that the authors shouldn’t use the 20 SNPs that they did. I only meant to suggest that the manuscript should justify how those 20 SNPs were selected.
7. The European Randomized Study of Screening for Prostate Cancer should be abbreviated ERSPC.
8. At one point USG is written as USA.
